# Towards Adapting Vision-Language Models for Semi-Supervised Domain Generalization

## Abstract

Semi-supervised Domain Generalization (SSDG) offers a cost-effective solution for generalizing models to unseen domains with limited labels. While existing SSDG methods, mainly built upon small-scale backbones, struggle to match fully supervised DG performance, large-scale vision-language models like CLIP have shown remarkable generalization through downstream fine-tuning. However, adapting these models to SSDG remains underexplored. In this paper, we identify a critical issue: existing popular fine-tuning methods suffer from under-utilizing unlabeled data in the semi-supervised learning frameworks, thereby overfitting the limited labeled data, leading to training collapse and generalization ability degradation. To address these challenges, we propose two novel components: (1) the De-False-Correlation Adapter (DFC-Adapter), which reduces false correlations to refine visual features and (2) Learnable Multi-granularity Text-guided Embedding Augmentation (LMTEA), which synthesizes semantic-aligned but domain-perturbed augmented visual embedding for consistency regularization through multi-granularity text guidance and learnable style encoding. Moreover, we establish the first-ever benchmark for CLIP fine-tuning methods in SSDG, conducting extensive experiments across six DG datasets and two ImageNet variants. Our results demonstrate that our method significantly outperforms existing CLIP fine-tuning approaches and achieves performance comparable to even fully supervised DG methods in some cases. Our code will be made public upon acceptance.

## 1 Introduction

Semi-supervised domain generalization (SSDG) aims to learn models that generalize to unseen domains by leveraging both limited labeled and abundant unlabeled data from multiple source domains. As the existing methods mainly build upon FixMatch Sohn et al. (2020) and its variants Zhang et al. (2021a); Wang et al. (2022a); Zhou et al. (2023); Qi et al. (2024b), they focus on three major technical directions of improving pseudo-labeling (PL) accuracy Zhang et al. (2021a); Zoha et al. (2024); Khan et al. (2024); Qi et al. (2024c), modeling domain-level information Galappaththige et al. (2024a); Wang et al. (2023b) and data-level consistency regularization Zhou et al. (2023). However, we notice the existence of three fundamental limitations: (1) Architectural constraints. Current approaches rely on small-scale backbones (e.g., ResNet-18 He et al. (2016)), which lack the scalability and generalization capacity of modern vision-language models (VLMs). (2) Pseudo-label reliability. Low PL accuracy injects noise into training, propagating errors and degrading generalization, which is a critical weakness given the limited labeled data in SSDG. (3) Augmentation poverty. Existing works Galappaththige et al. (2024b;a); Zhou et al. (2023); Qi et al. (2024b) mainly use simple predefined image-level augmentations, such as image rotation and flipping, hindering robustness to diverse domain shifts. Collectively, these limitations constrain further improvement of SSDG, precluding competitive performance against fully supervised DG methods.

Recent advances in adapting VLMs like CLIP Gao et al. (2024) through downstream fine-tuning Hu et al. (2022); Jia et al. (2022); Zhou et al. (2022) offer new opportunities. Popular fine-tuning methods, such as low-rank adaptation (LoRA) Hu et al. (2022) and prompt tuning techniques Jia et al. (2022); Zhou et al. (2022), offer promising pathways for customizing foundational models while preserving transferable knowledge. However, their application in SSDG scenarios remains underexplored. A critical limitation emerges here: existing downstream fine-tuning methods severely underutilize unlabeled data during SSDG training. As illustrated in Figure. 1, pseudo-

labeling confidence from CLIP's image-text pairing often falls below the hand-crafted confidence threshold, rendering large portions of unlabeled data inactive in gradient updates under the semi-supervised learning framework Sohn et al. (2020). This exacerbates the confirmation bias Arazo et al. (2020) inherent in semi-supervised learning, leading to overfitting on the limited labeled data and degrading generalization performance. Despite LoRA achieving high PL accuracy in the initial training period, only a small portion of unlabeled data contributes to gradient updates. Meanwhile, directly applying linear probing achieves marginal performance, but still degrades the pseudo-labeling accuracy due to potential overfitting. Another straightforward alternative, combining fine-tuning methods with a linear classifier, leads to training collapses similar to those of the vanilla fine-tuning. This suggests that preventing learnable modules from underutilizing the unlabeled data and overfitting to the limited labeled data is crucial for adapting VLMs to SSDG.

To address the challenges above and bridge the gap in VLMs research in SSDG, this paper proposes a De-False-Correlation Adapter (DFC-Adapter) and Learnable Multi-granularity Text-guided Embedding Augmentation (LMTEA) to prevent the confirmation bias from the perspective of architecture design and data augmentation. More specifically, DFC-Adapter learns generalizable knowledge to refine visual features in both spatial and semantic space by decreasing false correlations from both domain-specific biases and pre-trained knowledge. Meanwhile, LMTEA achieves a richer augmentation space for consistency learning with learnable style encoding and text-guided embedding augmentation from both object attribute level and global style level.

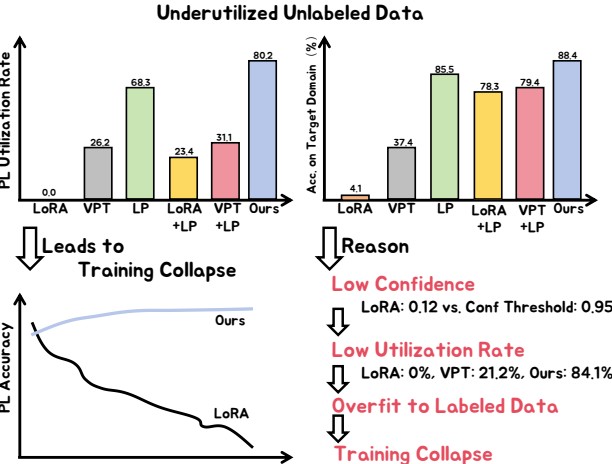

Figure 1: Illustration on the underutilized unlabeled data issue of existing fine-tuning methods (LoRA Hu et al. (2022) and VPT Jia et al. (2022)) under SSDG setting. LP is linear probing.

We evaluate our method on standard DG benchmark datasets and ImageNet variants, establishing the first benchmark for VLM fine-tuning methods in SSDG. Experimental results demonstrate that our method significantly outperforms existing methods and, in some cases, achieves performances comparable to fully supervised DG baselines. Our main contributions can be summarized as:

• We identify the low unlabeled data utilization in prior VLM fine-tuning methods under the SSDG scenarios, leading to training collapse and degradation of generalization performances.

• From the perspective of architecture design and data augmentation, we propose DFC-Adapter and LMTEA to adapt pre-trained VLMs to SSDG by preventing overfitting to the limited label data and training collapses caused by confirmation bias.

• Extensive experimental results across 8 datasets, where we achieve comparative performances with the fully supervised methods, demonstrate the superiority of the proposed method.

## 2 RELATED WORK

**Domain Generalization:** Domain generalization (DG) intends to learn a model from (multiple) source domains with transferable knowledge that can generalize to previously unseen target domains. Existing methods could be substantially categorized into data augmentation Li et al. (2021); Volpi et al. (2018); Xu et al. (2021); He et al.; Khan et al. (2021), domain alignment Hemati et al. (2023); Wang et al. (2022b), meta-learning Li et al. (2018); Chen et al. (2023) and optimization methods Cha et al. (2022); Yu et al. (2024); Wang et al. (2023a). Despite gaining performance improvements, the vast majority of the DG research is ill-equipped to process unlabeled data, largely

stemming from their foundational assumption of a fully supervised learning context. However, in real-world scenarios, it could be infeasible to curate a fully-labeled dataset with various domains for training, thereby limiting the applications of these fully supervised DG methods.

**Semi-supervised Domain Generalization:** Semi-supervised domain generalization (SSDG) has emerged as a promising avenue to address domain shift with limited labeled data. Existing SSDG methods Sohn et al. (2020); Zhou et al. (2023); Khan et al. (2024); Galappaththige et al. (2024a;b) are mainly developed from semi-supervised learning methods, such as FixMatch Sohn et al. (2020), demonstrating the crucial impact of pseudo-labeling (PL) and consistency regularization on SSDG performances. A handful of research Galappaththige et al. (2024b;a); Khan et al. (2024); Wang et al. (2023b) has been dedicated to improving PL accuracy during training. Wang et al. (2023b) approach domain-aware PL with a dual-classifier structure, while Galappaththige et al. (2024a) directly modulates the weight matrix of the classifier head with domain-level information. Meanwhile, on the aspect of consistency regularization, StyleMatch Zhou et al. (2023) introduces stochastic modeling on the classifier head and MultiMatch Qi et al. (2024b) formulates the multiple source domains into multiple local tasks and a global task for domain alignment. Despite demonstrating notable performance improvements, these methods significantly trail fully supervised DG performances. To the best of our knowledge, the vast majority of SSDG research exclusively applies small-scale convolutional networks as their backbones, limiting their further development. Meanwhile, for consistency learning, these methods depend on predefined image-level augmentation to generate augmented views in semi-supervised learning, which overlooks the rich semantic augmentation space.

**VLM Generalization:** Foundation vision-language models like CLIP Radford et al. (2021), pretrained on web-scale data, achieve strong out-of-distribution generation Shu et al. (2023). However, direct fine-tuning with task-specific data often harms robustness to distribution shifts Wortsman et al. (2022). To address this, parameter-efficient strategies, such as LoRA Hu et al. (2022), CoOP Zhou et al. (2022), VPT Jia et al. (2022), and MaPLE Khattak et al. (2023a) adapt only a few parameters while preserving generalizable knowledge. Beyond these, advanced methods further enhance OOD generalization via style-aware prompting Bose et al. (2024), disentangled representations Cheng et al. (2024), or knowledge distillation Addepalli et al. (2024). Despite these efforts, most fine-tuning methods assume fully supervised settings, leaving their behavior in semi-supervised domain generalization underexplored.. In parallel, feature augmentation approaches Dunlap et al. (2023); Qi et al. (2024a) enrich the training distribution by synthesizing semantically aligned but domain-perturbed features, enabling consistency regularization across domains. However, building upon the modality gap assumption Liang et al. (2022), they assume predefined text based on the domain name would be a perfect match of the visual features, which is not the case in real-world scenarios, thereby leading to potential semantically misaligned augmented features.

## 3 METHOD

### 3.1 PRELIMINARIES

**Problem Settings.** We denote each domain $d$ by $d = \{(x_i^d, y_i^d)\}_{i=1}^n$, where $x_i^d$, $y_i^d$ and $n$ is an input image, the corresponding ground-truth label and the total number of images in domain $d$, respectively. In the scenario of SSDG, there are only limited labeled samples, , each source domain has a labeled part $d^L = \{(x_i^d, y_i^d)\}$ and an unlabeled part $d^U = \{(u_i^d)\}$, where the number of samples in the unlabeled part is significantly higher than that in the labeled part. For the vision-language model CLIP Radford et al. (2021), we denote its image encoder and text encoder as $E_v$ and $E_t$, respectively.

**SSDG Pipeline.** We adopt FixMatch Sohn et al. (2020) as our baseline due to its empirical effectiveness in SSDG and conceptual simplicity. It integrates two SSL mechanisms. Pseudo-labeling (PL) assigns labels to unlabeled data when their maximum class probability exceeds a confidence threshold. Meanwhile, consistency regularization enforces prediction invariance across augmented views. FixMatch processes each sample in a mini-batch with both weak and strong augmentations. The total loss consists of: 1) supervised loss $\mathcal{L}_s$ applied on the weakly augmented version of the labeled data. And 2) $\mathcal{L}_u$ aligns predictions on strongly augmented unlabeled data with their pseudo-labels (generated from the weakly augmented versions). Despite its success in SSDG, FixMatch Sohn et al. (2020) faces significant challenges when adapted to downstream tasks with VLMs. As illustrated in Figure. 1, prevalent fine-tuning approaches fail to effectively utilize unlabeled data within

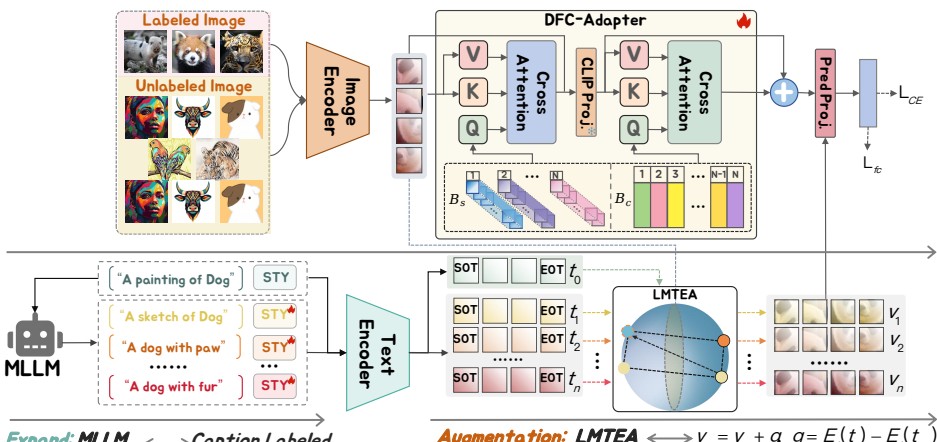

Figure 2: Overview of our proposed SSDG method, which introduces two key components of DFC-Adapter and LMTEA. $B_s$ and $B_c$ denote the spatial refinement bank and vision-language alignment correlation bank, respectively.

FixMatch's pseudo-labeling framework, resulting in both training collapse and generalization degradation.

## 3.2 METHOD OVERVIEW

As illustrated in Figure. 2, our framework comprises two key components: De-False-Correlation Adapter (DFC-Adapter) and Learnable Multi-granularity Text-guided Embedding Augmentation (LMTEA). Given visual features extracted by the image encoder $E_v$, the DFC-Adapter applies learnable knowledge banks as corrections to mitigate both domain-specific information and spurious correlations. During training, LMTEA synthesizes augmentation embeddings in the vision-language embedding space by incorporating learnable style encoding and external knowledge from large-language models (LLMs) to enhance consistency regularization.

## 3.3 DE-FALSE-CORRELATION ADAPTER LEARNING

CLIP's pretrained visual encoder $E_v$ often encodes spurious correlations between specific components, stemming from the source domains or pretrained knowledge, and semantic content Kim et al. (2024), leading to generalization degradation. To mitigate this while preserving cross-modal alignment and prevent overfitting, we propose the De-False-Correlation Adapter (DFC-Adapter) $P$, which dynamically suppresses biased feature activations through two sets of learnable knowledge bank: (1) a spatial refinement bank $B_s = \{b_{s,k}\}_{k=1}^K \in \mathbb{R}^{K \times d_s}$ that refines visual features over spatial tokens to enhance discriminative local patterns and (2) a semantic correlation alignment bank $B_c = \{b_{c,k}\}_{k=1}^K \in \mathbb{R}^{K \times d_c}$. $K$, $d_s$ and $d_c$ denote the number of learnable features in each bank and their corresponding feature dimensions, respectively. For a batch of input visual features $z = E_v(x) \in \mathbb{R}^{B \times L \times d_s}$, we first compute cross-attention between $B_s$ and $z$ to adaptively adjust spatial feature activations:

$$\begin{cases} P(z) = W_P(CA(z, b_s)), & CA(z, b_s) = \text{softmax}(\dfrac{Q_s K_s^T}{\sqrt{d}})V_s \\ Q_s = W_{Q,s}b_s, & K_s = W_{K,s}z, \quad V_s = W_{V,s}z, \end{cases} \tag{1}$$

where $W_{Q,s}$, $W_{K,s}$ and $W_{V,s}$ are the corresponding attention mapping matrix. $W_P$ denotes the mapping matrix for dimension alignment in the DFC-Adapter $P$. Then, we combine the output with the original visual feature in the vision-language alignment space:

$$z_s = \text{CLS}(z) + P(z), \tag{2}$$

where $\text{CLS}(\cdot)$ denotes projecting the class token to vision-language space. $z_s$ denotes the semantic visual feature in the vision-language embedding space. Subsequently, we incorporate the semantic correlation bank $B_c$ with the projected feature $z_s$ via cross-attention mechanism similar with that in Eq. 1 as:

$$z_f = z_f + CA(z_f, b_c), \tag{3}$$

where $z_f$ denotes the final feature for linear class prediction.

**De-False-Correlation Learning.** To mitigate domain-specific and pretrained knowledge biases while preserving semantic fidelity and cross-modality alignment, we propose a dual-level regularization mechanism. From the domain level, to decrease domain-specific bias information in the feature, the output of the DFC-Adapter should be varied across inputs from different domains and should be similar when inputs from the same source domain are given. Therefore, we formulate this as a contrastive domain alignment loss:

$$\mathcal{L}_{da} = -\frac{1}{n}\sum_{i=1}^{n}\log\frac{\sum_{i\neq j,d(z_i)=d(z_j),j=1,...,n}e^{-D(P(z_i),P(z_j))}}{\sum_{i\neq k,k=1,...,n}e^{-D(P(z_i),P(z_k))}}, \tag{4}$$

where $n$, $d(\cdot)$ and $D(\cdot)$ denote the batch size, obtaining the domain index from the source domains and L2 distance, respectively.

From the object-centric level, to mitigate spurious false correlations from the visual encoder of CLIP (deeming 'chair' and 'desk' identical), we propose to leverage the external knowledge from existing large-language models (LLMs). We prompt LLMs with a template to list commonly objects co-occurring with but essentially irrelevant to each class $c_{\text{src}}$. For example, $c_{\text{fc,chair}} = \{\text{desk}, \text{office}, ...\}$. Based on these counterfactual objects, we pull the visual feature of $c_{\text{src}}$ away from those text embeddings of $c_{fc}$, while maintaining image-text alignment with the original class text embedding as:

$$\mathcal{L}_{fc} = -\frac{1}{n}\sum_{i=1}^{n}\left(\sum_{j=1}^{|c_{fc,y_i}|}z_{f,i},E_t(c_{fc,y_j})) - \text{ITP}(z_{f,i},E_t(c_{src}))\right), \tag{5}$$

where $c_{fc,y_i}$ denote the false-correlation text list corresponding to class of $y_i$. ITP denotes calculating the cross-entropy loss on the image-text pairing prediction with the ground-truth label. The loss calculation object under the semi-supervised learning framework will be illustrated in detail in the latter section.

## 3.4 LEARNABLE MULTI-GRANULARITY TEXT-GUIDED EMBEDDING AUGMENTATION

To further enhance consistent regularization, we propose Learnable Multi-granularity Text-guided Embedding Augmentation (LMETA), which synthesizes semantic-aligned and domain-perturbed visual embeddings as augmentation samples during training. Unlike existing methods of VLM feature augmentation that rely on pre-defined domain names, LMETA introduces learnable style encoding and multi-granularity text-guided augmentation, effectively addressing semantic misalignment and enhancing the diversity of augmented features.

**Analysis on TEAM Qi et al. (2024a).** TEAM exploits the modality gap phenomenon Liang et al. (2022); Shi et al. (2023), which is the constant vector orthogonal to the span of image and text embeddings, to translate original visual features into novel domains. As shown in Figure. 3(a), the TEAM directly assumes that the text $t_{src}$ in the format of "a {*source domain name*} of the {*class name*}" would be the one match for the images from the source domain, and synthesize visual features from novel domains with another text $t_a$ "a {*augmented domain name*} of the {*class name*}" by:

$$v_a = v_x + g, \quad g = E_t(t_a) - E_t(t_{src}), \quad v_x = E_v(x) \tag{6}$$

However, the effectiveness of TEAM is based on the assumption that the predefined texts would be a well-matched pair for the image features. It fails when the domain name fails to represent the domain-specific information (domain name is set as the number of camera traps in TerraIncognita Beery et al. (2018)), leading to potential semantic misaligned augmented embedding and hindering consistency learning.

**Learnable Style Encoding.** To overcome this limitation, we propose to use the learnable style embedding to replace source domain name embedding. Inspired by previous work on image editing with diffusion models Gal et al. (2022); Zhang et al. (2023), where a learnable component is integrated with text embedding to generate novel concepts, we incorporate a set of learnable style components $s$. As shown in Figure. 3(b), given an image $x$ from the $d$-th domain, we select the corresponding style components to combine with the pre-defined text for noise prediction with the diffusion models. We optimize the learnable style component with the diffusion loss Rombach et al. (2022) to learn delicate domain-specific style components. To further improve the learning of

domain-specific information, we apply an orthogonality loss to reduce the correlation between each pair of style components:

$$\mathcal{L}_{\text{ortho}} = ||ss^T - \text{diag}(ss^T)||_2, \tag{7}$$

where diag means keeping only the diagonal entries. We combine the above two losses to learn the domain-specific style component before launching the fine-tuning of CLIP. With the learned style encoding, we combine them with the pre-defined text embedding to generate augmentation embeddings:

$$x_{aug} = v_x + \hat{g}, \quad \hat{g} = E_t(t_a) - \text{concat}(E_t(t_{src}) + s_d), \tag{8}$$

where concat means concatenate the text embeddings and $d$ denotes the domain index for selecting the corresponding style component.

**Multi-granularity Text-guided Augmentation.** To facilitate generalized learning at the object level, we propose a text-guided multi-granularity augmentation strategy based on Eq. 8, including both style-level and attribute-level augmentation. For style-level shifting, we prompt an LLM to generate a set of style words $t_{sty}$ and prepare the augmented text in the format of "*a {style word} of {class name}.*". Similarly to style-level shifting, for attribute-level augmentation, we prompt the LLM to generate the attribute sets for each class and craft the augmented text as "*a {class name} with {attribute}.*".

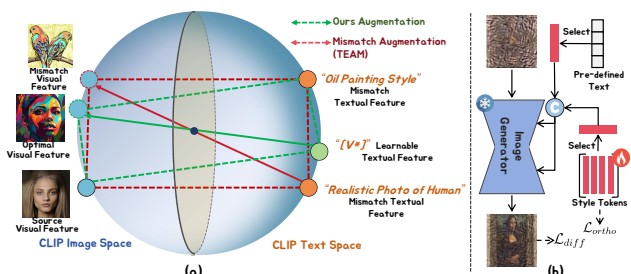

Figure 3: Illustration of (a) the misalignment issue of inconsistent modality gap from TEAM Qi et al. (2024a), and (b) the training process of the learnable style encoding in LMTEA.

### 3.5 TRAINING LOSS

Before fine-tuning the CLIP, we first train the domain-specific style encoding with a diffusion-based image generation model, as illustrated in the LMTEA section. Subsequently, we proceed to fine-tune the CLIP model with the DFC-Adapter. In the semi-supervised framework with both labeled and unlabeled data, the overall training loss is divided into supervised and unsupervised branches, following the FixMatch Sohn et al. (2020).

For labeled data, we compute cross-entropy losses for both original and augmented embeddings crafted by the proposed multi-granularity text-guided augmentation. And Jensen-Shannon divergence is applied to further enhance those embeddings consistency. Meanwhile, the domain-aware loss in Eq. 4, the false-correlation regularization in Eq. 5 are applied to the supervised data. Thereby, the supervised branch of the training loss is formulated as:

$$\begin{aligned}
\mathcal{L}_{sup} = & \mathcal{L}_{CE}(\hat{y}, y) + \mathcal{L}_{CE}(\hat{y}_{aug}, y) + \lambda_{JS}\mathcal{L}_{JS}(\hat{y}, \hat{y}_{aug}) \\
& + \lambda_{auxsup}(\mathcal{L}_{fc} + \mathcal{L}_{da}),
\end{aligned} \tag{9}$$

where $L_{CE}$, $\hat{y}$ and $\hat{y}_{aug}$ denote the cross-entropy loss, prediction on the original visual embedding and the augmented embedding of the labeled data, respectively.

For the unlabeled data, we first generate pseudo label for the unlabeled images by confidence threshold, obtaining a mask $m$ to select samples with high confidence predictions for loss calculation. A cross-entropy loss is then computed between the predictions of weakly and strongly augmented views, while a Jensen-Shannon divergence is applied between the weakly augmented view and the augmented embedding for the selected samples. Therefore, the unsupervised loss is formulated as:

$$\mathcal{L}_{unsup} = m(\mathcal{L}_{CE}(\hat{y}_w, \hat{y}_s) + \lambda_{JSu}\mathcal{L}_{JSu}(\hat{y}_w, \hat{y}_{aug})), \tag{10}$$

where $\hat{y}_w$ and $\hat{y}_s$ denote the predictions for the weakly and strongly augmented view. $\mathcal{L}_{JSu}$ represents Jensen-Shannon divergence loss calculated on the unsupervised data. $m$ is the mask for high confidence sample selection. Finally, the model is trained with the combination of $\mathcal{L}_{sup}$ and $\mathcal{L}_{unsup}$.

| Method | PACS | OfficeHome | 5 labels per class Digits | TerraInc. | VLCS | DomainNet | PACS | OfficeHome | 10 labels per class Digits | TerraInc. | VLCS | DomainNet |
|---|---|---|---|---|---|---|---|---|---|---|---|---|
| | | | | | *Fully Supervised* | | | | | | | |
| STYLIP | 98.05 | 84.63 | 81.38 | - | 86.94 | 62.02 | 98.05 | 84.63 | 81.38 | - | 86.94 | 62.02 |
| CoOp | 97.00 | 81.12 | 76.41 | - | 82.98 | 59.52 | 97.00 | 81.12 | 76.41 | - | 82.98 | 59.52 |
| VPT | 96.90 | 83.20 | - | 46.70 | 82.00 | 58.50 | 96.90 | 83.20 | - | 46.70 | 82.00 | 58.50 |
| FixMatch | | | | | *Semi-Supervised* | | | | | | | |
| +Linear Probe | 96.27 | 83.95 | 62.14 | 34.78 | 72.62 | 58.43 | 96.18 | 84.68 | 62.31 | 35.43 | 72.31 | 59.67 |
| +LoRA | 95.28 | 78.19 | 60.37 | 15.24 | 75.82 | 52.32 | 95.72 | 78.32 | 60.47 | 15.63 | 76.27 | 53.51 |
| +CoOP | 96.25 | 81.39 | 62.47 | 33.98 | 74.08 | 54.62 | 96.31 | 81.55 | 62.62 | 35.12 | 74.89 | 54.98 |
| +VPT | 96.19 | 80.33 | 61.86 | 35.64 | 76.42 | 49.81 | 96.23 | 80.54 | 62.43 | 35.56 | 76.53 | 50.42 |
| +CLIPood | 95.44 | 76.43 | 62.41 | 35.90 | 75.11 | 56.42 | 96.19 | 76.99 | 62.95 | 36.56 | 75.58 | 56.67 |
| +PromptSRC | 96.38 | 81.59 | 63.19 | 35.43 | 76.55 | - | 96.47 | 82.42 | 63.50 | 35.98 | 76.76 | - |
| +VL2V-SD | 95.46 | 85.47 | 66.40 | 38.42 | 76.74 | 57.92 | 95.68 | 86.12 | 66.98 | 38.96 | 77.83 | 58.41 |
| +MoA | 96.23 | 87.62 | 73.42 | 39.16 | 77.05 | **59.62** | 96.59 | 88.23 | 74.14 | 39.85 | 77.68 | 59.78 |
| +DGWM | 96.54 | 85.47 | 63.80 | 36.23 | 73.99 | 58.72 | 96.59 | 85.94 | 64.04 | 36.59 | 74.24 | 58.98 |
| +UPCSC | 96.68 | 86.32 | 69.84 | 37.04 | 77.49 | - | 96.78 | 87.21 | 73.66 | 38.48 | **78.25** | - |
| +Ours | **96.74** | **88.68** | **76.24** | **41.44** | **78.17** | 59.59 | **96.83** | **88.94** | **75.83** | **42.23** | 78.12 | **59.83** |

Table 1: Comparison with fine-tuning methods and SSDG SOTAs on popular DG benchmark datasets under the first setting. The best results under semi-supervised frameworks are in bold.

# 4 EXPERIMENTS

## 4.1 DATASETS, SETTINGS AND IMPLEMENTATION DETAILS

**Datasets.** We evaluate on standard DG benchmarks: PACS Li et al. (2017), Office-Home Venkateswara et al. (2017), Digits Zhou et al. (2020), TerraIncognita Beery et al. (2018), and VLCS Torralba & Efros (2011), plus the large-scale DG dataset DomainNet Peng et al. (2019). For scalability tests, we use semi-supervised ImageNet Deng et al. (2009) for training and evaluate on corrupted variants ( ImageNet-A Hendrycks et al. (2021b) and ImageNet-R Zhang et al. (2024)).

**Settings.** Two established SSDG settings Galappaththige et al. (2024b;a); Zhou et al. (2023); Wang et al. (2023b) are evaluated. (1) 5 or 10 labeled samples per class per source domain Galappaththige et al. (2024b;a). (2) One source domain is fully labeled, while the others are unlabeled Zhou et al. (2023); Wang et al. (2023b). The above two settings are referred to as the 1st and 2nd settings in the following text, respectively. For ImageNet-scale experiments, we fine-tune pretrained models with 1% labeled data. More details can be found in the supplementary material.

**Evaluation Protocol.** We follow the leave-one-domain-out protocol for evaluation. Following Galappaththige et al. (2024b;a), we report the average performances over 5 independent runs.

**Baselines.** We evaluate with: (1) standard downstream fine-tuning methods, including LoRA Hu et al. (2022), VPT Jia et al. (2022), CLIPood Shu et al. (2023), CoOP Zhou et al. (2022), Prompt-SRC Khattak et al. (2023b). (2) SOTA SSDG method UPCSC Lee et al. (2025) and DGWM Galappaththige et al. (2024a). (3) CLIP fine-tuning methods for supervised DG, such as VL2V-SD Addepalli et al. (2024) and MoA Lee et al. (2023). All methods are integrated with FixMatch Sohn et al. (2020). For reference, we also report fully supervised results of CLIP fine-tuning methods.

| Method (FixMatch) | PACS | OfficeHome | Digits | TerraInc. | VLCS | DomainNet |
|---|---|---|---|---|---|---|
| +Linear Probe | 96.41 | 84.52 | 57.04 | 32.69 | 66.05 | 49.79 |
| +LoRA | 88.59 | 73.24 | 52.88 | 14.03 | 62.31 | 48.34 |
| +CoOP | 92.46 | 82.43 | 56.70 | 29.35 | 58.94 | 49.05 |
| +VPT | 89.93 | 74.90 | 54.38 | 12.53 | 64.83 | 47.90 |
| +CLIPood | 89.25 | 75.21 | 55.63 | 28.04 | 63.13 | 51.43 |
| +PromptSRC | 94.24 | 79.42 | 57.84 | 31.53 | 64.08 | - |
| +MoA | 96.84 | 85.19 | 59.37 | 34.65 | 68.93 | 54.78 |
| +DGWM | 96.44 | 84.98 | 58.14 | 35.68 | 68.51 | 54.65 |
| +Ours | **96.89** | **86.73** | **64.89** | **39.59** | **72.52** | **54.88** |

Table 2: Comparison on popular DG benchmark datasets under the second SSDG setting. The best results are labeled in bold.

## 4.2 MAIN RESULTS

**Results on standard DG benchmarks.** Tables. 1 and 2 demonstrate our method's superior performances across DG benchmarks. We significantly outperform fine-tuning methods (LoRA Hu et al. (2022), CoOp Zhou et al. (2022), VPT Jia et al. (2022)), which show substantial degradation when

Table 3: Comparison with PEFT methods and SSDG SOTAs on ImageNet variants.

| Method(FixMatch) | ImageNet-A | ImageNet-R |
|---|---|---|
| +Linear Probe | 72.34 | 50.34 |
| +LoRA | 66.74 | 45.62 |
| +VPT | 71.80 | 49.95 |
| +CLIPood | 73.87 | 52.54 |
| +MoA | 77.61 | 56.74 |
| +Ours | **79.05** | **62.49** |

Table 4: Ablation study of our method. Linear probing is referred to as the initial setting.

| DFC-Adapter | LMTEA | OfficeHome |
|---|---|---|
| ✗ | ✗ | 84.68 |
| Adapter-only | ✗ | 86.12(+1.44) |
| w/o $\mathcal{L}_{da}$ | ✗ | 86.74(+2.06) |
| w/o $\mathcal{L}_{fc}$ | ✗ | 86.57(+1.89) |
| ✓ | ✗ | 87.22(+2.54) |
| ✓ | w/o LSE | 87.93(+3.25) |
| ✓ | Attr. Only | 88.13(+3.45) |
| ✓ | Sty. Only | 88.46(+3.78) |
| ✓ | ✓ | **88.94**(+4.26) |

(a) Training evolution

(b) Hyperparameter study.

Figure 4: **(a)** Evolution of the pseudo-label (PL) accuracy and unlabeled data utilization rate during training on OfficeHome Venkateswara et al. (2017). **(b)** Hyperparameter study on the learnable banks' sizes in DFC-Adapter and the auxiliary losses weight $\lambda_{auxsup}$.

adapted to SSDG settings due to limited unlabeled data utilization. Notably, our approach exceeds robust fine-tuning baseline MoA Lee et al. (2023) by 2.28% and 2.38% on TerraIncognita in Setting 1 and rivals fully supervised SOTA methods on PACS/OfficeHome. Further surpassing SSDG SOTA DGWM Galappaththige et al. (2024a), we achieve 0.87% and 0.85% gains on DomainNet under both settings.

**Results on ImageNet variants.** Table. 3 validates the efficiency of the proposed method on corrupted ImageNet variants, where we outperform the previous robust fine-tuning SOTA MoA Lee et al. (2023) by 1.44% and 5.65% on ImageNet-A Hendrycks et al. (2021b) and ImageNet-R Hendrycks et al. (2021a), respectively. This pronounced improvement under ImageNet-R's challenging distribution shifts confirms the superiority of LMTEA in simulating visual embeddings from different unseen domains.

### 4.3 ABLATION STUDY

**Contributions of each component.** We evaluate key component contributions through ablation studies in Table. 4. As shown, the DFC-Adapter alone improves linear probing performances by 2.54%, demonstrating its effectiveness in refining representations. Conversely, removing our learnable style encoding causes significant degradation, confirming its essential role in generating semantically-aligned augmented embeddings for domain generalization.

**Training evolution.** We present the evolution of the PL utilization rate and PL accuracy between LoRA Hu et al. (2022) and our approach throughout the training process in Figure. 4(a). As LoRA continually fails to adapt more unlabeled data for training, it overfits the limited labeled data, leading to degradation of the PL accuracy. However, after the initial training period, our method uses a growing number of unlabeled data for training, leading to high PL accuracy and effective training.

**Hyperparameter Studies.** We conduct hyperparameter analysis on the size of the knowledge banks, where $K_s$ and $K_c$ define the size of the spatial refinement bank and semantic alignment bank, as well as the weights of the auxiliary loss weight $\lambda_{auxsup}$, on the Office-Home dataset Venkateswara et al. (2017). As shown in Figure. 4(b), when we set a moderate size for the learnable banks, the performances remain stable overall, indicating that our method is not sensitive to the bank size generally. However, with an excessive bank size, the performances undergo significant degradation, as the bank with such a size would not be trained thoroughly and inject potential noise during inference. In terms of the weight for the auxiliary loss, the changes in the weight merely affect the performance in a small margin.

**Effectiveness of feature refinement from the Learnable Knowledge Banks.** To validate the effectiveness of the learnable knowledge banks on feature refinement, we provide t-SNE visualizations of the process of the feature changes when passing through the DFC-Adapter. As

Table 5: Comparisons with LoRa on OfficeHome with different confidence thresholds for PL.

| Conf. Threshold | LoRA | Ours |
|---|---|---|
| 0.65 | 72.47 | 84.05 |
| 0.85 | 77.84 | 88.46 |
| 0.95 | **78.19** | **88.68** |

Table 6: Comparisons with different adapter designs Gao et al. (2024); Zhang et al. (2021b)

| Method | PACS | OH | VLCS | TI |
|---|---|---|---|---|
| CLIP-Adapter | 96.23 | 86.15 | 78.12 | 36.06 |
| Tip-Adapter | 96.36 | 86.40 | 77.51 | 35.88 |
| Ours | **96.83** | **88.94** | **78.30** | **39.23** |

Table 7: Comparison with different embedding augmentation methods.

| Method | PACS | OH | VLCS | TI |
|---|---|---|---|---|
| LADS | 95.87 | 87.41 | 77.89 | 36.08 |
| TEAM | 96.61 | 88.46 | 78.22 | 37.65 |
| Ours | **96.83** | **88.94** | **78.30** | **39.23** |

Table 8: Comparison of DFC-Adapter with different adapter designs Gao et al. (2024); Zhang et al. (2021b)

| Method | PACS | OfficeHome | VLCS | TerraInc. |
|---|---|---|---|---|
| CLIP-Adapter | 96.23 | 86.15 | 78.12 | 36.06 |
| Tip-Adapter | 96.36 | 86.40 | 77.51 | 35.88 |
| Ours | **96.83** | **88.94** | **78.30** | **39.23** |

Table 9: Comparison with different text-guided embedding augmentation methods and variants of our LMTEA.

| Method | PACS | OfficeHome | VLCS | TerraInc. |
|---|---|---|---|---|
| LADS | 95.87 | 87.41 | 77.89 | 36.08 |
| TEAM | 96.61 | 88.46 | 78.22 | 37.65 |
| Ours | **96.83** | **88.94** | **78.30** | **39.23** |

shown in Figure. 5, after interacting with the two sets of knowledge banks, the features become better clustered and more discriminative than the original CLIP features, demonstrating that the learned knowledge banks can refine the visual features for better performance.

**Confidence Distribution Analysis.** In this section, we analyze whether a lower confidence threshold could improve previous PEFT methods. The confidence of image-text pairing predictions from CLIP is significantly lower than the confidence threshold for pseudo-labeling, leading to a low utilization rate of the unlabeled data. Therefore, an intuitive approach would be to lower the confidence threshold to include more unlabeled data. However, as shown in Tab. 5, lowering the confidence threshold cannot lead to better performance, as it may introduce noisy labels during training, thereby degrading generalization.

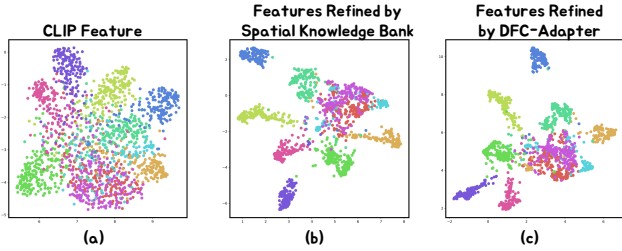

Figure 5: t-SNE visualizations of features processed before and after the learnable knowledge banks in DFC-Adapter.

**Comparisons with different adapters.** In Table. 8, to demonstrate the superiority of the DFC-Adapter, we compare it with different adapter designs, including CLIP-Adapter Gao et al. (2024) and Tip-Adapter Zhang et al. (2021b). Notably, our method outperforms CLIP-Adapter and Tip-Adapter by margins of 3.17% and 3.35% on TerraIncognita Beery et al. (2018).

**Comparisons with different embedding augmentation methods.** We compare LMTEA with the most related method TEAM Qi et al. (2024a) and LADS Dunlap et al. (2023) in Table. 9. LMTEA outperforms TEAM by a large margin of 1.58% on TerraIncognita Beery et al. (2018), where the domain-specific style information of the dataset is hard to describe by text, demonstrating the effectiveness of our learnable style encoding for mining domain-specific style information.

## 5 CONCLUSION

In this paper, we address the challenge of adapting vision-language models (VLMs) for semi-supervised domain generalization (SSDG). We reveal that existing downstream fine-tuning methods suffer from low utilization rates of unlabeled data, leading to overfitting and degraded generalization. To tackle this, we propose DFC-Adapter and LMTEA, which aim to prevent the confirmation bias of semi-supervised learning from both perspectives of architecture design and data augmentation. Our method significantly outperforms prior methods, achieving comparative results to fully supervised methods on several datasets. This work establishes the first benchmark for VLM SSDG, highlighting the potential of adapting VLMs for robust generalization under limited labeled data.

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

# Appendix of "Towards Adapting Vision-Language Models for Semi-Supervised Domain Generalization"

**The Use of Large Language Models (LLMs).** Large Language Models (LLMs) were used to aid or polish the writing of this manuscript. Specifically, we used Claude-4-Sonnet solely for language polishing and grammatical refinement of the written text. All research contributions, including the main ideas, technical approaches, experimental work, and scientific insights presented in this paper, are entirely the work of the human authors. The LLM usage is limited to improving the clarity and readability of the already-written content without altering the substance or meaning of our work.

## A    REPRODUCIBILITY STATEMENT

To ensure reproducibility, we have made the following efforts: (1) We will release our code and models. (2) We provide experiments setup and implementation configurations. (3) We elaborate on our evaluation protocol in detail. We believe these measures will enable other researchers to reproduce our work and further advance the field.

**Performance in fully-supervised DG setting.** Notably, the main component of the proposed DFC-Adapter and LMTEA could also be applied in the fully-supervised setting. We thus provide experiment results in Tab. 10. As seen, our method achieves comparative performances, even in fully supervised DG settings, compared with DG SOTA STYLIP Bose et al. (2024).

| Method | PACS | OfficeHome | TerraInc. | VLCS | DomainNet |
|--------|------|-----------|-----------|------|-----------|
| STYLIP Bose et al. (2024) | 98.05 | 84.63 | - | 86.94 | 62.02 |
| Ours | 97.48 | 88.72 | 45.96 | 84.72 | 62.14 |

Table 10: Experiment results under fully-supervised DG settings.

**Integrated with different SSL baselines.** Since our method could be integrated with different SSL baselines, we provide results in Tab. 11, demonstrating the wide applicability of the proposed method.

| Method | PACS | OfficeHome | TerraInc. |
|--------|------|-----------|-----------|
| FixMatch Sohn et al. (2020) | 97.48 | 88.72 | 45.96 |
| FlexMatch Zhang et al. (2021a) | 95.42 | 85.04 | - |
| StyleMatch Zhou et al. (2023) | 97.52 | 88.52 | 46.32 |

Table 11: Experiment results when integrating our method with different SSL baselines. Notably, integrating with FixMatch Sohn et al. (2020) is used as the default setting in the main paper.

## B    IMPLEMENTATION DETAILS

ViT-B/16 version of the pre-trained CLIP Radford et al. (2021) is used as our backbone in all the experiments. For all the compared baselines, we follow the same sets of hyper-parameters from their DG ones, except for VPT Jia et al. (2022) and LoRA Hu et al. (2022). The learning rate is set as 1e-2 for VPT and LoRA. The learning rate of our method is set as 1e-3. We train all the models for 10 epochs across all the datasets. In the SSDG evaluation, we use the splits of the datasets from previous SSDG SOTA DGWM Galappaththige et al. (2024a). The confidence threshold for pseudo-labeling is set as 0.95.

