# OpenReview forum: "Towards Adapting Vision-Language Models for Semi-Supervised Domain Generalization"
_ICLR.cc/2026/Conference — ICLR 2026 Conference Withdrawn Submission_

### Official Review · Reviewer_Gx2X · 2025-10-30

**Soundness:** 2
**Presentation:** 3
**Contribution:** 3
**Rating:** 4
**Confidence:** 4

**Summary:**

The paper addresses semi-supervised domain generalization (SSDG) using vision-language models (VLMs) such as CLIP, highlighting that existing fine-tuning methods (LoRA, VPT, etc.) underutilize unlabeled data, causing overfitting and training collapse. To overcome this, the authors introduce: DFC-Adapter and LMTEA. Experiments across multiple DG datasets (PACS, OfficeHome, VLCS, DomainNet, ImageNet variants) show better performance.

**Strengths:**

1. The paper is well organized and easy to follow.

2. The experimental part is comprehensive, test on several DG dataset and ImgeNet.

**Weaknesses:**

1. The motivation of the paper is not very clear to me. Comparing the proposed method only to LoRA may not be entirely fair — it would be more appropriate to compare against a semi-supervised baseline such as FixMatch. The key question is not merely whether unlabeled data are used, but how effectively they are utilized. How is this utilization quantified? For example, if a method applies consistency regularization to all unlabeled samples, does that imply 100% utilization? Clarifying this definition and its measurement would make the motivation and evaluation more convincing.

2. In Tables 8 and 9, it appears that the experiments are conducted under a standard domain generalization (DG) setting rather than a semi-supervised one — is that correct? If so, a more meaningful comparison would be combinations such as CLIP-Adapter + LMTEA, DFC-Adapter + LADS, or TIP-Adapter + TEAM, to better isolate the contribution of each component.

3. Following the above point, the experimental results suggest that the proposed approach performs well not only in semi-supervised settings but also in fully supervised and potentially even unsupervised DG contexts. This raises a concern about the motivation: what specifically motivated the design of the adapter and the augmentation components? Were they primarily introduced to enhance the utilization of unlabeled data, or to improve general fine-tuning performance of VLMs? The paper should clarify how these components are conceptually and practically tied to semi-supervised training, rather than general adaptation.

**Questions:**

1. Could you please provide more details on how the pseudo-label utilization rate is calculated for different methods such as LP or VPT? Specifically, how is “utilization” defined and measured in practice, for instance, does it correspond to the proportion of unlabeled samples that contribute to gradient updates, or some other metric?

2. Could you clarify how the spatial refinement bank and the semantic correlation bank function in practice? It would be helpful to understand their initialization strategy and whether they effectively act as a form of single-layer attention. Additionally, could you visualize or provide examples illustrating how these banks refine the features during training?

3. Table 6 and 7 are the same as Table 8 and Table 9.

---

### Official Review · Reviewer_LoRW · 2025-10-31

**Soundness:** 2
**Presentation:** 1
**Contribution:** 2
**Rating:** 2
**Confidence:** 4

**Summary:**

In this paper, the authors discuss the underutilization of unlabeled data in semi-supervised domain generalization. To address, they propose DFC-Adapter to mitigate false correlations of visual features. Meanwhile, the LMTEA module generates domain-perturbed augmented visual embeddings to facilitate consistency regularization.

**Strengths:**

1.	The DFC-Adapter and LMTEA modules enhance the model’s ability to effectively utilize unlabeled data.
2.	The authors conduct evaluations on a diverse and comprehensive set of datasets.

**Weaknesses:**

1.	The paper lacks rigor in several aspects, for example:
+ Tables 6–9 contain duplicated content and inconsistent sizing.
+ In Figure 3(b), the term L_diff lacks a formal mathematical definition.
+ In Equation (5), the first term is missing the ITP component

2.	The proposed approach is based on CLIP for Single-Source Domain Generalization (SSDG). Therefore, the experimental comparison should prioritize CLIP-based SSDG or few-shot DG methods to ensure fair evaluation and highlight the effectiveness of the proposed design.

3.	Table 1 shows that the proposed method yields only limited performance improvement, particularly on the large-scale DomainNet dataset.

4.  Considering that on the ImageNet variants, ImageNet-A and ImageNet-R have smaller category spaces than the source domain ImageNet, it might be more appropriate to include an additional experiment on ImageNet-Sketch, which shares the same category space as the source domain.

**Questions:**

As shown in Figure 4, the PL utilization rate of LoRA remains at zero throughout, which seems abnormal.

---

### Official Review · Reviewer_JSyp · 2025-10-31

**Soundness:** 3
**Presentation:** 3
**Contribution:** 2
**Rating:** 4
**Confidence:** 4

**Summary:**

In this paper, the authors propose two components for utilizing unlabeled data during semi-supervised domain generalization training. Specifically, De-False-Correlation Adapter (DFC-Adapter) tries to reduce false correlations from domain-specific biases and pre-trained knowledge to refine visual features. And Learnable Multi-granularity Text-guided Embedding Augmentation (LMTEA) synthesizes domain-perturbed augmented visual embeddings for consistency regularization.

**Strengths:**

1.	This paper proposes two components, including DFC-Adapter and LMTEA, to adapt pre-trained VLMs to SSDG without overfitting.

2.	The proposed methods improved the utilization rate and accuracy of unlabeled data a lot.

**Weaknesses:**

1.	The details of Figure 5 are unclear. What do the different colors represent - domains or categories? Specifically, does each cluster correspond to samples from the same category across different domains, or samples from the same domain across different categories?

2.	The manuscript requires more rigorous proofreading and editing. In particular:
(a) Tables 6 and 8 appear to be identical, and Tables 7 and 9 are also identical.
(b) Line 136 contains a typographical error: "underexplored.." (duplicate punctuation).

3.	Given the strong pretrained knowledge of CLIP, it would be more convincing to include its zero-shot performance as a reference in the experimental comparison. For instance, on DomainNet with CLIP ViT-B/16, the reported zero-shot accuracy is around 57.8% [1,2], suggesting that the results in Table 2 may not be directly comparable.

4.	Additionally, since the proposed method leverages LLMs to augment domain and attribute prompts, it would be appropriate to include zero-shot CLIP results with the same prompt augmentation as an additional baseline.

5.	The paper states that Equation (6) holds "based on the assumption that the predefined texts would be a well-matched pair for the image features." However, in Equation (8), the augmentation follows the same structure, but E(ta) may also not be well-matched to the target image features either. As a result, the augmented embeddings might not be optimal.

[1] PromptStyler: Prompt-driven Style Generation for Source-free Domain Generalization.ICCV 23

[2] UMFC: Unsupervised Multi-Domain Feature Calibration for Vision-Language Models. NeurIPS 24

**Questions:**

See Weakness part

---

### Official Review · Reviewer_UAE2 · 2025-11-01

**Soundness:** 3
**Presentation:** 3
**Contribution:** 3
**Rating:** 4
**Confidence:** 5

**Summary:**

This paper studies the Semi-Supervised Domain Generalization (SSDG) using large Vision-Language Models (VLMs) like CLIP. The authors highlight a key limitation in existing VLM fine-tuning methods, i.e., limited utilization of unlabeled data, leading to overfitting on limited labeled data and training collapse. As such, the authors propose a De-False-Correlation Adapter (DFC-Adapter) to refines visual features & reduce spurious correlations, and a Learnable Multi-granularity Text-guided Embedding Augmentation (LMTEA) that generates diverse, semantic-aligned feature augmentations. The method is integrated into a FixMatch SSL framework, evaluated on a self-established benchmark, and achieves consistent performance gain.

**Strengths:**

- Study the feature disentanglement in the context of foundation models.
- Propose a new benchmark.
- Qualitative visualization is provided.

**Weaknesses:**

- My primary concern is the validity and relevance of the experimental setup. The proposed benchmark, while a contribution, is primarily a recombination of existing 2D recognition datasets. These tasks are relatively simple and may have been part of the model's pre-training data, raising doubts about whether the results demonstrate *real* generalization. Furthermore, the cited literature to justify this setup is outdated, with most references predating 2023. Thus, I am not sure whether the benchmark can connect to contemporary, real-world challenges.
- The proposed method rely on external models for generating false correlation samples and style encodings, which introduces much complexity and computational overhead. Furthermore, the paper does not analyze the sensitivity of the results to the choice of these external models. In other words, as those images may have been used for many complicated training objective, why should we still consider evaluating them for image classification?
- The performance gain highlighted in Table 1 is limited compared with the other SOTA methods.
- Definition of domain-specific bias: A critical issue in domain generalization is to clear define the domain specific and domain agnostic component for the network training. Currently, the pipeline in effect first defines domain-specific elements and then disentangle them to get the domain-agnostic (generalizable) ones, which may not be intuitive and even introduce some noise hurting the effectiveness.

**Questions:**

Please see the comments in weakness part.

---

### Note · Authors · 2026-01-15

I have read and agree with the venue's withdrawal policy on behalf of myself and my co-authors.